# Reduced neutralisation of the Delta (B.1.617.2) SARS-CoV-2 variant of concern following vaccination

**Chris Davis[1], Nicola Logan[1], Grace Tyson[1], Richard Orton[1], William T. Harvey[1], Jonathan S. Perkins[2], Guy Mollett[2], Rachel M. Blacow[2], The COVID-19 Genomics UK (COG-UK) Consortium, Thomas P. Peacock[3], Wendy S. Barclay[3], Peter Cherepanov[4], Massimo Palmarini[1], Pablo R. Murcia[1], Arvind H. Patel[1], David L. Robertson[1], John Haughney[2], Emma C. Thomson[1,5]\*, Brian J. Willett[1]\*, on behalf of the COVID-19 DeplOyed VaccinE (DOVE) Cohort Study investigators**

**1** MRC-University of Glasgow Centre for Virus Research, University of Glasgow, Glasgow, United Kingdom, **2** Clinical Research Facility, Queen Elizabeth University Hospital, Glasgow, United Kingdom, **3** Department of Infectious Disease, Imperial College London, London, United Kingdom, **4** The Francis Crick Institute, London, United Kingdom, **5** Department of Clinical Research, London School of Hygiene and Tropical Medicine, London, United Kingdom

\* emma.thomson@glasgow.ac.uk(ET); brian.willett@glasgow.ac.uk(BW)

**Data Availability Statement:** All relevant experimental data are within the manuscript and its Supporting Information files. Patient details have

## Abstract

Vaccines are proving to be highly effective in controlling hospitalisation and deaths associated with SARS-CoV-2 infection but the emergence of viral variants with novel antigenic profiles threatens to diminish their efficacy. Assessment of the ability of sera from vaccine recipients to neutralise SARS-CoV-2 variants will inform the success of strategies for minimising COVID19 cases and the design of effective antigenic formulations. Here, we examine the sensitivity of variants of concern (VOCs) representative of the B.1.617.1 and B.1.617.2 (first associated with infections in India) and B.1.351 (first associated with infection in South Africa) lineages of SARS-CoV-2 to neutralisation by sera from individuals vaccinated with the BNT162b2 (Pfizer/BioNTech) and ChAdOx1 (Oxford/AstraZeneca) vaccines. Across all vaccinated individuals, the spike glycoproteins from B.1.617.1 and B.1.617.2 conferred reductions in neutralisation of 4.31 and 5.11-fold respectively. The reduction seen with the B.1.617.2 lineage approached that conferred by the glycoprotein from B.1.351 (South African) variant (6.29-fold reduction) that is known to be associated with reduced vaccine efficacy. Neutralising antibody titres elicited by vaccination with two doses of BNT162b2 were significantly higher than those elicited by vaccination with two doses of ChAdOx1. Fold decreases in the magnitude of neutralisation titre following two doses of BNT162b2, conferred reductions in titre of 7.77, 11.30 and 9.56-fold respectively to B.1.617.1, B.1.617.2 and B.1.351 pseudoviruses, the reduction in neutralisation of the delta variant B.1.617.2 surpassing that of B.1.351. Fold changes in those vaccinated with two doses of ChAdOx1 were 0.69, 4.01 and 1.48 respectively. The accumulation of mutations in these VOCs, and others, demonstrate the quantifiable risk of antigenic drift and subsequent reduction in vaccine efficacy. Accordingly, booster vaccines based on updated variants are likely to be required over time to prevent productive infection. This study also suggests that

been anonymised in keeping with relevant clinical ethics approvals.

**Funding:** The COVID-19 DeplOyed VaccinE (DOVE) study is funded by the Medical Research Council core award (MC UU 1201412; M.P) We acknowledge the support of the G2P-UK National Virology Consortium (MR/W005611/1) funded by the UKRI (M.P.,E.C.T.). COG-UK is supported by funding from the Medical Research Council (MRC) part of UK Research & Innovation (UKRI), the National Institute of Health Research (NIHR) and Genome Research Limited, operating as the Wellcome Sanger Institute (G.M., R.M.B., D.L.R., E. C.T.). W.T.H. is funded by the MRC (MR/R024758/ 1). N.L. and B.J.W. were funded by the Biotechnology and Biological Sciences Research Council (BBSRC, BB/R004250/1), G.T. was funded by the Department of Health and Social Care (DHSC, BB/R019843/1). The funders had no role in study design, data collection and analysis, decision to publish, or preparation of the manuscript. We are indebted to Therese McSorley for recruiting participants to the DOVE study. We thank all the researchers who have shared genome data openly via the GISAID Initiative.

**Competing interests:** The authors have declared that no competing interests exist

two dose regimes of vaccine are required for maximal BNT162b2 and ChAdOx1-induced immunity.

## Author summary

The SARS-CoV-2 virus is likely to have emerged following a cross-species jump from bats to humans in Wuhan, China at the end of 2019. As the virus has evolved to adapt to humans, new viral variants have emerged that incorporate mutations in the "spike" gene. The spike protein is expressed on the surface of the virus, it facilitates entry into human cells and is the main target of neutralising antibodies. Mutations change the shape of the spike protein, preventing antibody recognition and enabling the virus to escape from the immunity induced by vaccination. Using samples collected from vaccinated people as part of the COVID-19 Deployed Vaccine Cohort Study (DOVE), we assessed the capacity of different variants (beta, kappa and delta) to evade the protective immune response in recipients of the Astra Zeneca ChAdOx1 and Pfizer BNT162b2 vaccines, both of which are based on the early Wuhan virus spike gene. We noted a reduction in neutralisation of both the beta and delta variants by vaccine sera from recipients of both vaccines. While vaccines remain highly effective at preventing severe infection and death, ongoing monitoring of vaccine effectiveness is indicated as the virus continues to evolve over time, especially in vulnerable groups.

## Introduction

The B.1.617.2 (Delta) variant that spread from India in March 2021 is now the dominant SARS-CoV-2 variant type in the United Kingdom [1], replacing the B.1.1.7 (Alpha; "Kent") variant and spreading rapidly across the globe. The B.1.617.2 variant has been introduced into the UK on multiple occasions, most commonly associated with international travel from India where it has caused a large wave of COVID-19 infections and placed unprecedented demand on healthcare services [2]. A key component of the UK response to COVID-19 is a campaign of mass vaccination, prioritizing the population by age and other risk groups. Vaccination began in December 2020 using the BNT162b2 mRNA vaccine (PfizerBioNTech). The ChAdOx1 adenovirus vectored vaccine (Oxford-AstraZeneca) was added from January 2021, with the mRNA-1273 vaccine (Moderna) available from April 2021. Priority was given to administering the first dose of vaccine to as much of the UK population as possible, with second doses given within 12 weeks, in line with the guidance of the Joint Committee on Vaccination and Immunisation (JCVI). This delayed dosing strategy is now being challenged by the emergence of the B.1.617.2 lineage of SARS-CoV-2. Recent data from Public Health England suggest that following exposure to this lineage, effectiveness of the BNT162b2 vaccine is reduced to 33.5% after one dose, and 87.9% following two doses [3]. Further, the two-dose effectiveness of the ChAdOX1 vaccine is reduced to 59.8% following exposure to B.1.617.2 [3].

The early virus sequences detected in India were reported to have two key amino acid substitutions (L452R and E484Q) in the receptor-binding domain of the spike glycoprotein, the main immunodominant region and the region involved in ACE2 binding. Accordingly, this resulted in the widespread usage of the "*double mutant*" misnomer, and initial designation as the B.1.617 Pango lineage. Availability of further sequence data led to the assignment of sub-lineages: B.1.617.1, B.1.617.2 and B.1.617.3, of which B.1.617.2 is now the dominant variant in

the UK. The three lineages are characterized by the spike mutation L452R, whilst E484Q is present in B.1.617.1 and B.1.617.3 but not B.1.617.2. The substitution L452R has been shown previously to reduce binding by several monoclonal antibodies [4,5,6,7,8] and convalescent plasma [6]. Globally, L452R has emerged independently in several lineages since November/December 2020 suggesting a role in immune-evasion and/or virus adaptation [9]. L452R is one of the defining mutations of the lineage B.1.427/B.1.429, a variant of interest (VOI) first identified in California and associated with reduced neutralisation titres with plasma from vaccinated or convalescent individuals [7]. Investigation of the effect of RBD mutations on binding of convalescent plasma by deep mutational scanning suggests the impact of E484Q is similar to that of E484K [10], which has been shown widely to diminish antibody binding, including those elicited by vaccination [8,11].

In this study, we investigated the neutralising capacity of sera from participants in the COVID-19 DeplOyed VaccinE (DOVE) Cohort Study who had been vaccinated with the BNT162b2 mRNA vaccine (Pfizer-BioNTech) or the ChAdOx1 adenovirus-vectored vaccine (Oxford-AstraZeneca) as part of the national deployed vaccine strategy.

## Materials and methods

### Ethics statement

All participants gave written informed consent to take part in the study. The study was approved by the North-West Liverpool Central Research Ethics Committee (REC reference 21/NW/0073).

### Serum samples

Serum samples were collected from healthy volunteers participating in the COVID-19 Deployed Vaccine Cohort Study (DOVE), a cross-sectional cohort study to determine the immunogenicity of deployed COVID-19 vaccines against evolving SARS-CoV-2 variants. DOVE is a post-licensing cross-sectional cohort study of individuals vaccinated with deployed vaccines as part of the UK response to the COVID-19 pandemic. Adult volunteers aged at least 18 years, were recruited into the observational study at 14 days post first or second dose of vaccine.

### Preparation of SARS-CoV-2 antigens for ELISA

S1 and RBD antigens were prepared as described previously [12]. Briefly, the SARS-CoV-2 RBD and S1 constructs, spanning SARS-CoV-2 S (UniProt ID P59594) residues 319–541 (RVQPT. . .KCVNF) and 1–530 (MFVFL. . .GPKKS), respectively, were produced with C-terminal twin Strep tags in the mammalian expression vector pQ-3C-2xStrep38. A signal peptide from immunoglobulin kappa gene product (METDTLLLWVLLLWVPGSTGD) was used to direct secretion of the RBD construct. Proteins were produced by transient expression in Expi293F cells growing in FreeStyle 293 medium. Conditioned media containing secreted proteins were harvested at two timepoints, 3–4 and 6–8 days post-transfection. Twin Strep-tagged proteins were captured on Streptactin XT (IBA LifeSciences), eluted, and then purified to homogeneity by size exclusion chromatography through Superdex 200 (GE Healthcare). Purified SARS CoV2 antigens, concentrated to 1–5 mg/ml by ultrafiltration were aliquoted and snap-frozen in liquid nitrogen prior to storage at -80˚C.

## ELISA for SARS-CoV-2 antibodies

ELISAs for SARS-CoV-2 antibodies were performed as described previously [13]. Briefly, 96-well plates were coated overnight at 4˚C with purified SARS-CoV-2 antigens in phosphate-buffered saline (PBS). Wells were blocked for 1 hr at room temperature in blocking buffer consisting of PBS with 0.05% Tween 20 (PBS/Tween) and 1X casein (Vector labs., Peterborough, UK). Plates were then washed 3x in PBS/Tween prior to incubation with 50μL of each serum sample diluted 1:100 in blocking buffer. Each plate included two pooled negative controls and two pooled positive controls. Sera were incubated for 1 hour at room temperature. Plates were then washed 3x with PBS/Tween, before incubation for 1 hour with horseradish peroxidase (HRP)-conjugated rabbit anti-human IgG (Bethyl labs., Cambridge Bioscience, Cambridge, UK) diluted 1:2500 in blocking buffer. Plates were washed a further 3x in PBS/Tween before addition of the 3,3′,5,5′-tetramethylbenzidine (TMB) liquid substrate (Sigma Aldrich, Merck, Dorset, UK). Colour development was allowed to proceed for 10 minutes before the addition of 1M $H_2SO_4$ stop solution, at which point the absorbance was determined at 450nm on a Multiskan FC plate reader. Full validation of the S1 and RBD ELISA has been described previously [13].

## Measurement of neutralising antibody activity using viral pseudotypes

HEK293, HEK293T, and 293-ACE2 cells were maintained in Dulbecco's modified Eagle's medium (DMEM) supplemented with 10% foetal bovine serum, 200mM L-glutamine, 100μg/ml streptomycin and 100 IU/ml penicillin. HEK293T cells were transfected with the appropriate SARS-CoV-2 S gene expression vector (wild type or variant) in conjunction with p8.91 [14] and pCSFLW [15] using polyethylenimine (PEI, Polysciences, Warrington, USA). HIV (SARS-CoV-2) pseudotypes containing supernatants were harvested 48 hours post-transfection, aliquoted and frozen at -80˚C prior to use. The SARS-CoV-2 spike glycoprotein expression construct for Wuhan-Hu-1 was obtained from Nigel Temperton, University of Kent. The S gene of B.1.351 (South Africa) was based on the codon-optimised sequence of the Wuhan-Hu-1 expression construct, synthesised by Genscript (Netherlands) and sub-cloned into the pCDNA6 expression vector. S gene constructs bearing the B.1.617.1 and B.1.617.2 S genes were based on the codon-optimised spike sequence of SARS-CoV-2 [16] and generated using the QuikChange Multi Site-Directed Mutagenesis Kit (Agilent, USA). Constructs bore the following mutations relative to the Wuhan-Hu-1 sequence (GenBank: MN908947): **B.1.351** – D80A, D215G, L241-243del, K417N, E484K, N501Y, D614G, A701V; **B.1.617.1** –T95I, G142D, E154K, L452R, E484Q, D614G, P681R, Q1071H; **B.1.617.2** –T19R, G142D, E156del, F157del, R158G, L452R, T478K, D614G, P681R, D950N. 293-ACE2 target cells [13] were maintained in complete DMEM supplemented with 2μg/ml puromycin.

Neutralising activity in each sample was measured by a serial dilution approach. Each sample was serially diluted in triplicate from 1:50 to 1:36450 in complete DMEM prior to incubation with HIV (SARS-CoV-2) pseudotypes, incubated for 1 hour, and plated onto 239-ACE2 target cells. After 48–72 hours, luciferase activity was quantified by the addition of Steadylite Plus chemiluminescence substrate and analysis on a Perkin Elmer EnSight multimode plate reader (Perkin Elmer, Beaconsfield, UK). Antibody titre was then estimated by interpolating the point at which infectivity had been reduced to 90% of the value for the no serum control samples.

## Results

### Characterisation of B.1.617.2 spike sequences

The B.1.617.2 lineage has spread rapidly across the globe following detection in India in late 2020. According to GISAID (https://www.gisaid.org - accessed on 10/06/2021), a total of 31,997 sequences (Europe = 24,606, Asia = 4,974, North America = 2,210, Oceania = 163, Africa = 36, South America = 8) have been assigned to lineage B.1.617.2, predominantly from the UK (n = 22,619; reflecting the large-scale UK sequencing effort). The first B.1.617.2 sequence in the UK occurred on the 18th March 2021 when the dominant UK lineage was B.1.1.7, and since the end of May 2021, B.1.617.2 accounts for the majority of SARS-CoV-2 samples sequenced (Fig 1A). In order to make sure that our available reagents matched the majority of the circulating B.1.617.2 variants, we assessed the relative frequency of each spike mutation in all the available sequences (Fig 1B). Amino acid substitutions T19R, G142D, R158G, L452R, T478K, D614G, P681R, D950N and deletion Δ156–157 were present in the majority of the B.1.617.2 variants as chosen in the spike constructs used in our assays described below. The B.1.617.2 lineage continues to evolve, acquiring new mutations of concern such as K417N in the sub-lineage AY.1/B.1.617.2.1 (Fig 1B).

Although each Pango lineage has a distinct mutation set, there are several similarities between the spike mutational profiles of the VOCs B.1.351, B.1.1.7 and B.1.617.2 (Fig 2). They each have a deletion within the N-terminal domain supersite (NTDSS), at least one mutation in the receptor binding motif (RBM), and B.1.1.7 and B.1.617.2 each have a mutation at P681 within the furin cleavage site.

### Antibody response post-vaccination

Sera were collected from 156 healthy individuals who had received one dose (n = 37) or two doses (n = 50) of BNT162b2 (Pfizer-BioNTech), or one dose (n = 50) or two doses (n = 18) of ChAdOx1 (Oxford/AstraZeneca) vaccines. Samples were screened initially by ELISA for reactivity with recombinant S1, RBD and N from the Wuhan-Hu-1 SARS-CoV-2 sequence. Of those individuals vaccinated with BNT162b2, only one individual given a single dose (1/37) failed to mount a detectable antibody response against S1, all other samples were positive for reactivity against both S1 and RBD (Fig 3A). In contrast, four individuals given a single dose (4/50) of ChAdOx1 failed to react with S1, although two of these samples bound the RBD antigen (Fig 3B). All samples from individuals immunised with two doses of either BNT162b2 or ChAdOx1 reacted strongly against both S1 and RBD. Antibody reactivity (A450nm) was significantly higher following the second dose of either BNT162b2 (S1 and RBD, p<0.0001) or ChAdOx1 (S1 p = 0.0006; RBD p = 0.0014) compared with a single dose of the respective vaccines. Moreover, reactivity against S1 was significantly greater in the groups immunised with either one (p = 0.0152) or two (p = 0.0145) doses of BNT162b2 in comparison with the groups immunised with one or two doses of ChAdOx1 respectively. Similarly, reactivity against RBD was higher in samples from the groups immunised with either one (p = 0.0029) or two (p = 0.0018) doses of BNT162b2 in comparison with one or two doses of ChAdOx1 respectively. Eight individuals were identified with reactivity against SARS-CoV-2 N suggesting prior, undocumented exposure to SARS-CoV-2 or a related coronavirus. Exclusion of samples from these individuals did not affect the analyses (S1 Table).

### Neutralising antibody response

Neutralising antibodies were measured against HIV(SARS-CoV-2) pseudotypes expressing spike glycoproteins from either the vaccine sequence (Wuhan-Hu-1), or the variants B.1.617.1,

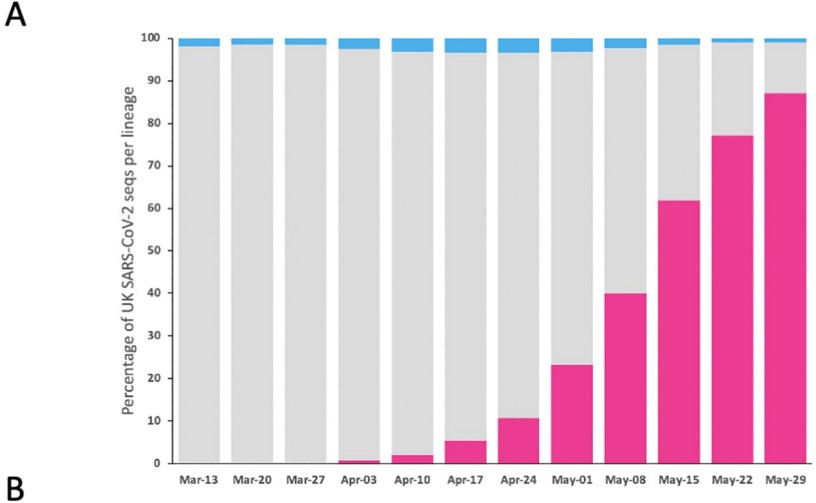

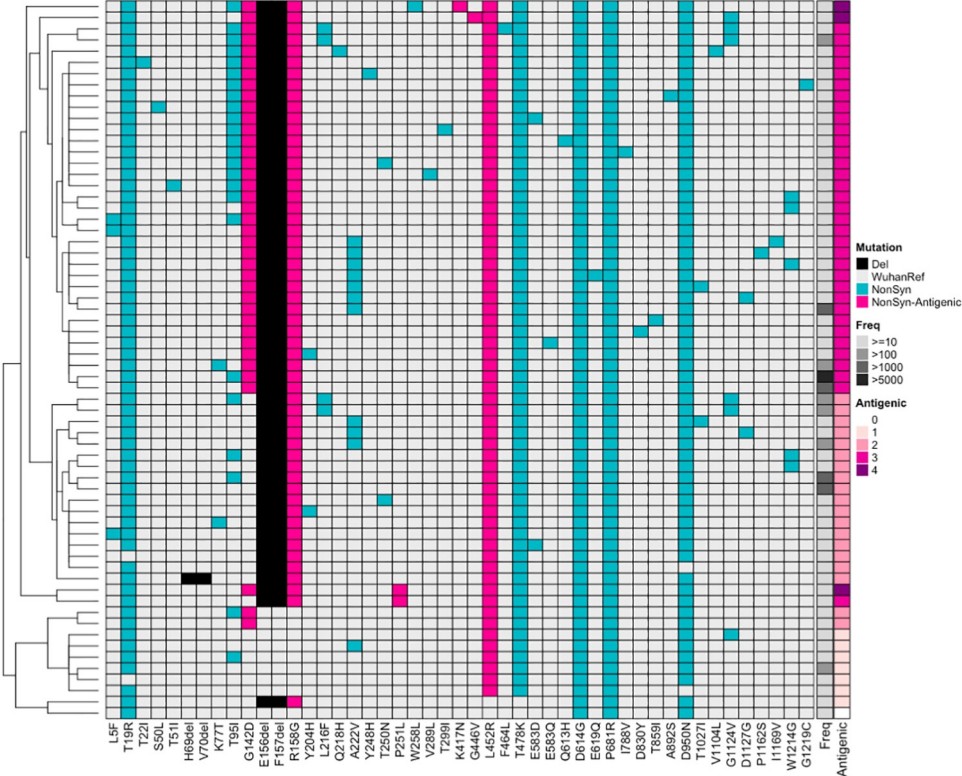

**Fig 1. Emergence of B.1.617.2 in the UK.** (A). Weekly SARS-CoV-2 genome sequences of delta/B.1.617.2 (pink), alpha/B.1.1.7 (grey), and all other lineages (blue) in the UK, represented as a (stacked) percentage of all UK sequences that week, up to the week beginning 29[th] May 2021. **Heatmap visualisation of spike mutations within UK B.1.617.2 SARS-CoV-2 genome sequences** (B). Columns represent different amino acid mutations within the spike protein, whilst rows represent different specific combinations of spike mutations ("backbones"). Only non-synonymous mutations (blue or pink for those with a known antigenic effect) and deletions (black) were considered, and only backbones observed 10 or more times are displayed. The observed frequency for each backbone is visualised in the Freq column whilst the antigenic column represents the total number of known antigenic mutations in the backbone; the backbone from the AY.1 lineage (derived from Nepal; containing mutations W258L and K417N) is also included (top row). The heatmap is hierarchically clustered based on the Euclidean distance between spike backbones (rows); backbones missing specific mutations/deletions could be indicative of Ns (failed amplicons) in the genome sequence at those sites rather than true absence.

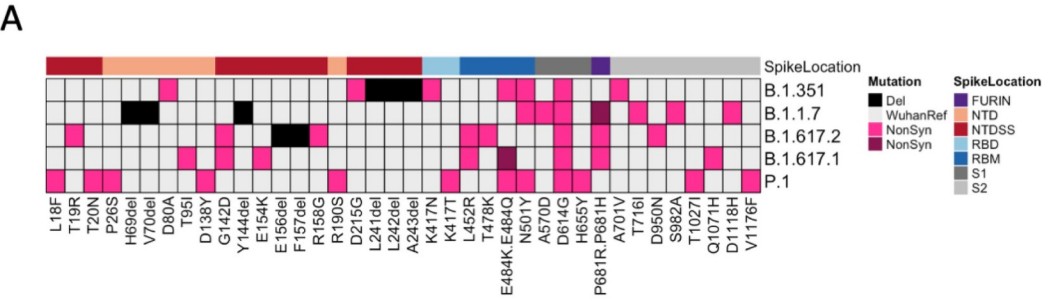

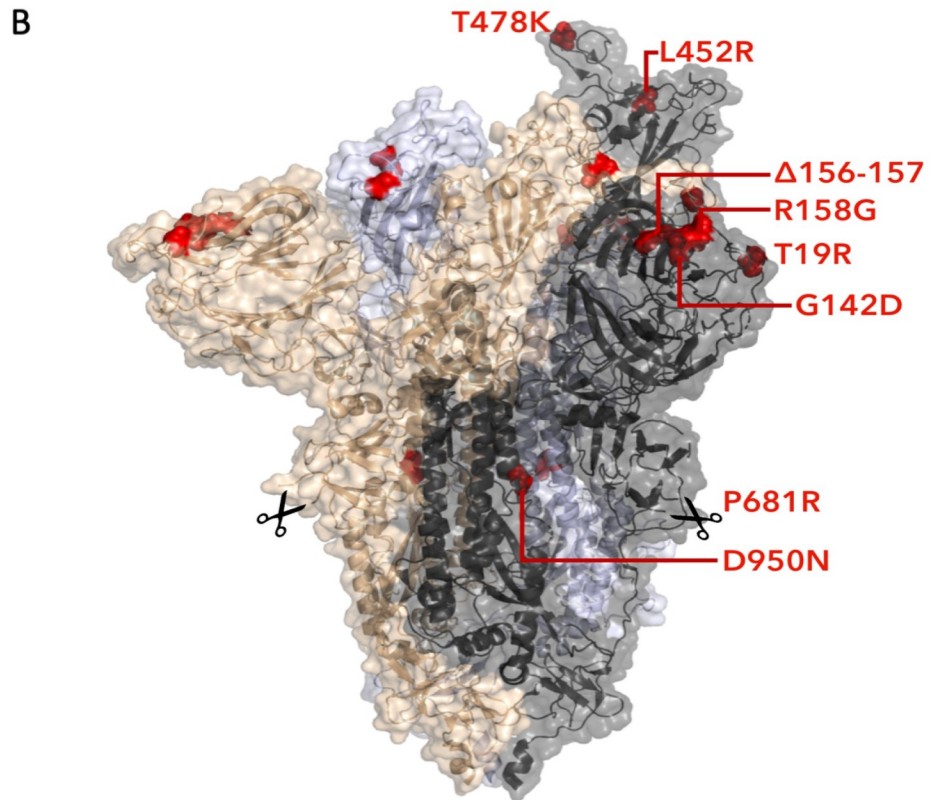

**Fig 2. Spike mutations within variants of concern.** (A) Non-synonymous mutations (pink) and deletions (black) are shown for the variants of concern: B.1.1.7, B.1.351, B.1.617.1 and B.1.617.2, and P.1. Purple is used to distinguish secondary non-synonymous mutations at the same position, for example E484K (pink) and E484Q (purple). The region of the spike protein the mutation is located is highlighted on the top row; N-terminal domain (NTD), NTD antigenic supersite (NTDSS), receptor binding domain (RBD), receptor binding motif (RBM), furin cleavage site, S1 (NTD, NTDSS, RBD, RBM and furin are also in S1) and S2 subunits. **Spike protein structure showing key B.1.617.2 mutations** (B). Surface representation of spike homotrimer in open conformation with one upright RBD overlaid with ribbon representation (RCSB Protein Data Bank ID 6ZGG [26], with different monomers shown in black, pale blue and gold. Residues involved in B.1.617.2 lineage defining substitutions or deletions are shown as red spheres on each of the three monomers and are labelled on the monomer with an upright RBD, shown in black. Scissors mark the approximate location of an exposed loop (residues 677–688), containing the furin cleavage site, and including residue 681, which is absent from the structure.

B.1.617.2 and B.1.351. Antibody titres were estimated by interpolating the point at which

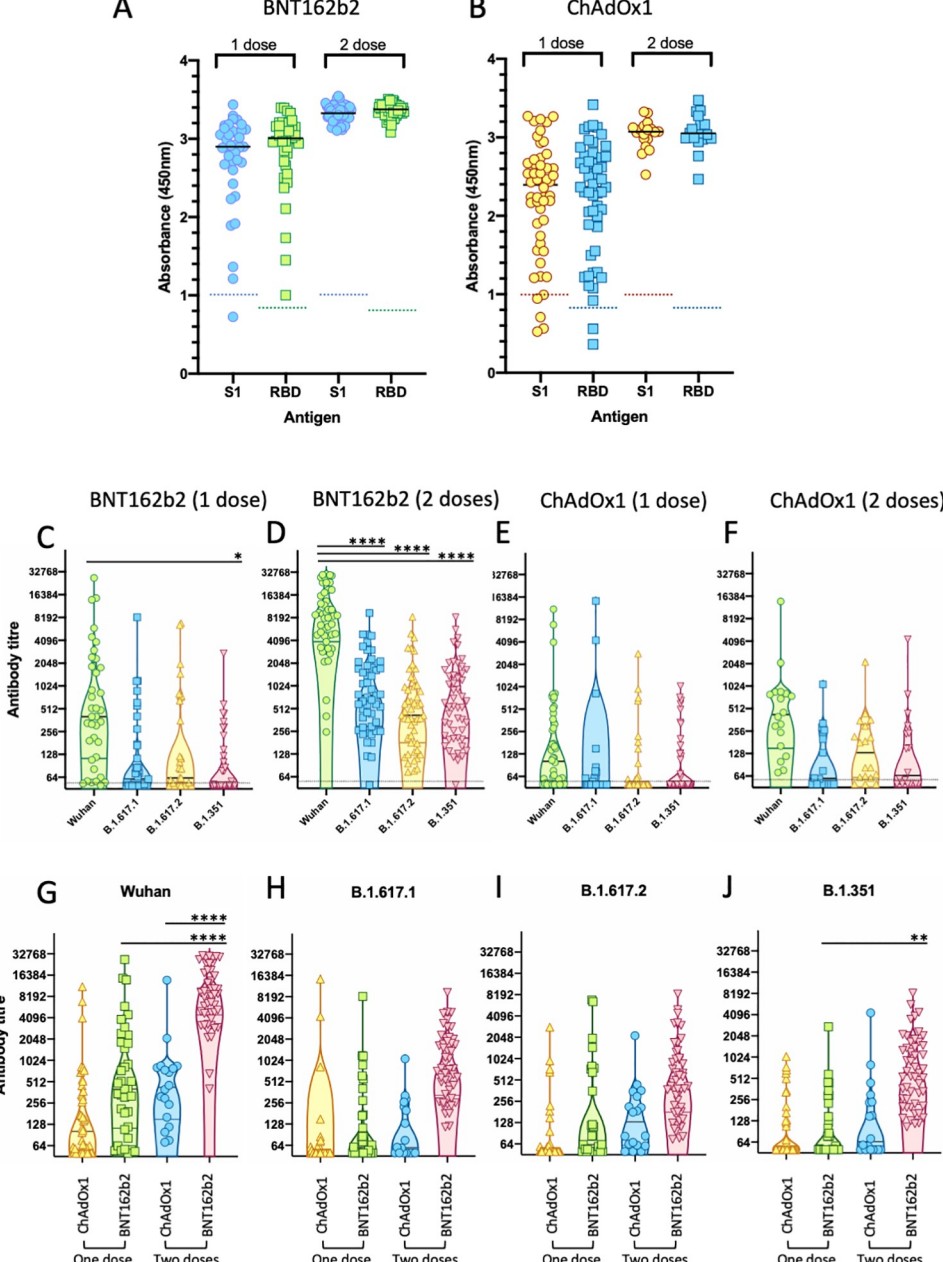

**Fig 3. Antibody response elicited by SARS-CoV-2 vaccines.** Sera from participants in the DOVE study were analysed by ELISA or pseudotype-based neutralisation assay. (A, B) ELISA reactivity in sera from individuals vaccinated with one or two doses of either BNT162b2 (A) or ChAdOx1 (B) was measured against recombinant S1 and RBD. Each point represents A450nm, cut-offs for S1 and RBD respectively are denoted by dotted lines. (C-F) Neutralising antibodies from individuals vaccinated with one or two doses of BNT162b2 (C and D) or ChAdOx1 (E and F) were quantified against HIV(SARS-CoV-2) pseudotypes bearing the Wuhan-Hu-1, B.1.617.1, B.1.617.2 or B.1.351 spike glycoproteins. Each point is the mean of three replicates, violin plot illustrates median plus quartiles. Mean titres were compared by one-way ANOVA. (G-J) Neutralising antibody titres were categorised based into the four viral variants against which they were determined; Wuhan-Hu-1 (G), B.1.617.1 (H), B.1.617.2 (I) or B.1.351 (J). Samples were then subdivided into one or two doses of BNT162b2 or ChAdOx1 respectively for each variant and compared (one-way ANOVA, Tukey's multiple comparison's test). Vaccination with two doses of BNT162b2 induced significantly higher titres of antibody against the Wuhan-Hu-1 virus than either one dose of BNT162b2, or two doses of ChAdOx1 (**** p<0.0001, ** p = 0.0011).

infectivity (luciferase activity) was reduced by 50%. Neutralizing antibodies were induced by vaccination with both the BNT162b2 (Fig 3C and 3D) and ChAdOx1 (Fig 3E and 3F) vaccines and two doses of either vaccine boosted the titre of neutralizing antibodies. Antibody titres were greatest against the Wuhan-Hu-1 spike glycoprotein and lower against the variants B.1.617.1, B.1.617.2 or B.1.351. Samples from all individuals vaccinated with two doses of BNT162b2 neutralised Wuhan-Hu-1 efficiently (mean titre = 11473, n = 50), however, mean antibody titres against the variants B.1.671.1, B.1.617.2 and B.1.351 were reduced by 7.77-fold, 11.30-fold and 9.56-fold respectively (significant, p<0.0001) (S2 Table). The mean antibody titre induced by vaccination with two doses of ChAdOx1 (mean titre = 1325.6, n = 18) was significantly lower than that induced by two doses of BNT162b2 (mean titre = 11473) (Fig 3G and S2 Table). After a single dose of ChAdOx1, 17 of 50 of vaccinated individuals (34%) had antibody titres ≤50. In comparison, only 5 of 37 individuals (13.5%) vaccinated with a single dose of BNT162b2 had antibody titres ≤50. These data are consistent with ChAdOx1 inducing a weaker antibody response than BNT162b2 following primary immunisation. Comparison of antibody responses measured by ELISA and by pseudotype-based neutralisation assay (S1 Fig) illustrated the marked difference between the responses elicited by one or two doses of vaccine. After the first dose of either BNT162b2 or ChAdOx1, antibody responses varied greatly, from very low (negative in both assay formats) to high (positive in both formats). Two doses of BNT162b2 elicited higher antibody titres than two doses of ChAdOx1. Stratification of the data into three groups based on titre (<50; 50 to 500; and >500) (S2 Fig) illustrated the reduction in neutralisation of the variants; only the samples from individuals given two doses of BNT162b2 retained measurable neutralising activity against the three variants, B.1.617.1, B.1.617.2 and B.1.351. After two doses of ChAdOx1, all samples displayed neutralising activity against Wuhan-hu-1, however only 55.6% neutralised B.1.617.1, 83.4% neutralised B.1.617.2, and 61.1% neutralised B.1.351.

The mean titre of antibodies detected in individuals vaccinated with BNT162b2 against all the VOCs analysed was higher than those present in sera from individuals vaccinated with ChAdOx1 (Fig 3G–3J). Vaccination with two doses of BNT162b2 induced significantly higher neutralising antibody titres against the Wuhan-Hu-1 virus than either one dose of BNT162b2, or two doses of ChAdOx1. Although the mean neutralising antibody titres against the variants were lower than those against the Wuhan-hu-1 vaccine strain, titres against Wuhan-hu-1 correlated broadly with the cross-neutralising titres against the variants B.1.617.1, B.1.617.2 and B.1.351 (S3 Fig and S3 Table).

When the age distribution of the study cohorts was compared, it was notable that the participants vaccinated with the ChAdOx1 vaccine were on average 15 years older than those vaccinated with BNT162b2 (43 versus 58 respectively; S2 Table and S4 Fig), consistent with the shifting governmental policy on age-group targeting mid-study. Insufficient samples were available from younger participants (ChAdOx1) or older participants (BNT162b2) to examine the effects of age on antibody response.

## Discussion

The Delta variant B.1.617.2 that originated in India has rapidly become the dominant lineage in the UK. This variant is characterised by mutations in the genome that are associated with immune escape in other SARS-CoV-2 lineages. In this study, we aimed to investigate the neutralisation profile of sera from participants in the DOVE deployed vaccine cohort study against B.1.617 sub-lineage variants. We compared neutralisation of B.1.617 variants with the original Wuhan-Hu-1 virus that has been used as the prototype for all currently deployed vaccines and the B.1.351 variant that originated in South Africa. The B.1.351 variant has been shown to be

associated with reduced neutralisation and breakthrough infection in clinical trials [17]. We aimed to quantify neutralisation profiles from sera obtained from recipients of the BNT162b2 and ChAdOx1 vaccines after one or two doses of vaccine, informing the UK strategy of maximising first dose rollout of vaccination in the population.

Our study showed that using the HIV (SARS-CoV-2) pseudotype-based system, neutralisation of the B.1.617.1, B.1.617.2 and B.1.351 variants was significantly lower in magnitude in comparison with the Wuhan-Hu-1 variant in participants vaccinated either with BNT162b2 or ChAdOx1, while two doses of BNT162b2 induced significantly higher neutralizing antibody titres against the Wuhan-Hu-1 and B.1.351 variants than one dose. Previous studies looked at the antibody resistance of B.1.351 and noted a reduction in sensitivity of 10.3-fold and 12.4-fold in sera from BNT162b2 (Pfizer)-vaccinated and mRNA-1273 (Moderna)-vaccinated individuals respectively, using a vesicular stomatitis virus (VSV) pseudotype-based assay [8] while live virus-based microneutralisation assays revealed a reduction of 8 to 14-fold in the neutralisation of B.1.351 by convalescent plasma [18]. Despite using distinct technical approaches to measuring neutralising antibody responses, the reductions in magnitude observed in our study (9.56-fold for B.1.351) and prior studies [8,18] are broadly similar, hence we can have confidence in concomitant estimates of the magnitude of neutralisation escape by variants B.1.617.1 and B.1.617.2. A recent study using sera from individuals vaccinated with BNT162b2 employed a high-throughput, live virus microneutralisation assay and observed a more modest reduction in neutralisation of 4.9-fold for B.1.351 and 5.8-fold for B.1.617.2 [19], perhaps reflecting subtle differences in the parameters being measured by live virus based assays using virus cultured in Vero E6 cells (viral entry, replication and spread) and pseudotype based assays (viral entry and transfer vector gene expression only). More modest escape from neutralisation by vaccine sera by B.1.617.1 (2.7-fold for BNT162b2) and B.1.617.2 (2.5-fold for BNT162b2) was noted by Liu et al. [20]. This more modest effect may reflect a combination of variables: the reference strain was SARS-CoV-2/human/AUS/VIC01/ 2020 (S247R) and not Wuhan-hu-1, with both the Victoria and B.1.617.2 (A222V) viruses grown in Vero cells. Further, the BNT162b2 vaccine sera came from a compressed schedule study (4–14 days after 2 doses 3 weeks apart), while BNT162b2 vaccinated individuals in our study were given 2 doses 9–10 weeks apart. Finally, BNT162b2 titres against Victoria and B.1.617.2 were estimated using a focus reduction neutralisation test (FRNT) [20] as opposed to a pseudotype-based neutralisation assay in this study.

It was notable that both our study and Wall *et al* [19] used sera from "real world" vaccinated individuals rather than clinical trial participants and both showed a significant increase in neutralisation after two vaccine doses. In contrast, recent data from 20 sera collected from clinical trial participants vaccinated with BNT162b2 showed relatively similar levels of neutralising antibodies against B.1.617.1, B.1.617.2, B.1.618 (all first identified in India) and B.1.525 (first identified in Nigeria) using a live virus assay (plaque reduction assay) [21]. In that study, geometric mean plaque reduction neutralization titres against the variant viruses appeared lower than the titre against USA-WA1/2020, an early Wuhan-Hu-1-like virus, however all sera tested neutralized the variant viruses at titres of at least 40 and displayed very uniform titres against each variant (albeit titres for B.1.617.1 were somewhat lower). Those results contrast with the spread of neutralising antibody levels across variants observed in our study and in Wall *et al* [19]. These discrepancies may be due to the source and number of the sera analysed or to the methodology used.

The mean titre of antibodies detected in individuals vaccinated with BNT162b2 against all the VOCs analysed was higher than those present in sera from individuals vaccinated with ChAdOx1. Vaccination with two doses of BNT162b2 induced significantly higher neutralising antibody titres against the Wuhan-Hu-1 virus than either one dose of BNT162b2, or two doses

of ChAdOx1. Further, the mean antibody titre induced by vaccination with two doses of ChAdOx1 was significantly lower than that induced by two doses of BNT162b2. Levels of neutralising antibody detected post-vaccination correlate strongly with the degree of protection from infection afforded [22,23]. Hence the differences we observed in levels of neutralising antibody elicited by BNT162b2 and ChAdOx1, and the reductions in cross-neutralisation against the VOCs in comparison with the vaccine strain Wuhan-hu-1, suggest that the degree of immunity afforded by vaccination will vary depending on the variants currently circulating in the community and the vaccines employed. Continuous surveillance of neutralising responses against VOCs will provide a valuable tool for predicting the likely efficacy of vaccines in curtailing the spread of novel variants.

Due to vaccines being used in batches targeted at decreasing age groups in the UK, comparisons between neutralisation responses in recipients of the ChAdOx1 versus the BNT162b2, vaccine responses may also be affected by age differences between these groups [19,24]. For example, in this study, participants vaccinated with the ChAdOx1 were on average 15 years older than those vaccinated with BNT162b2. Hence the effects of age will need further investigation as samples from broader populations of age-matched individuals become available.

In summary, we found that the B.1.617.2 variant, currently dominant in the UK is associated with significantly reduced neutralisation from vaccine sera obtained from recipients of the BNT162b2 or ChAdOx1 vaccines. Neutralisation titres were higher following two doses of vaccine. These data are in keeping with recent vaccine effectiveness studies published by Public Health England (PHE) and Public Health Scotland (PHS), in which test negative case control designs were used to estimate the effectiveness of vaccination against symptomatic disease [3,25]. In the PHE study, data on all symptomatic sequenced cases of COVID-19 in England was used to estimate the proportion of cases with B.1.617.2 compared to the preceding B.1.1.7 variant by vaccination status. Effectiveness was found to be lower after one dose of vaccine with B.1.617.2 (33.5%) compared to B.1.1.7 (51.1%), with similar results for both vaccines. After two doses of BNT162b2 vaccine, effectiveness reduced from 93.4% with B.1.1.7 to 87.9% with B.1.617.2. Following two doses of ChAdOx1, effectiveness reduced from 66.1% with B.1.1.7 to 59.8% with B.1.617.2. In addition, sequenced cases detected after one or two doses of vaccination had a higher odd of infection with B.1.617.2 compared to unvaccinated cases (OR 1.40; 95%CI: 1.13–1.75). The PHS data from the EAVE-2 study employed S gene dropout status (a non-exclusive marker of the B.1.1.7 lineage but not the B.1.617.2 lineage) to estimate vaccine effectiveness. The BNT162b2 vaccine was found to be 92% in the *S* gene-negative group (inferred as B.1.1.7) versus 79% in the *S* gene-positive group (inferred as B.1.617.2). The ChAdOx1 vaccine was reduced from 73% in *S* gene-negative cases versus 60% in *S* gene-positive ones. These data and ours suggest that pseudotype-based neutralisation assays are likely to reveal correlates of immunity to SARS-CoV-2 virus variants and further investigation to correlate neutralisation titres with vaccine failure is warranted.

The UK strategy for prioritisation of one-dose vaccination of the population with a second dose within 12 weeks is strongly associated with a significant reduction in deaths and hospitalisation associated with COVID-19 infection. However, the emergence of the B.1.617.2 variant (or others with similar neutralisation profiles, such as B.1.351) may necessitate a modified approach, to counter the increase in infections observed with the B.1.617.2 variant in the UK. More positively, despite the lower humoral response observed, hospitalisation rates of vaccinated people remain low. This indicates that vaccine-elicited immune responses can moderate disease severity even in the face of a reduction in the antibody response. High transmission rates of the B.1.617.2 variant in single-dose vaccine recipients or those previously infected with another variant may risk the evolution of vaccine-adapted variants. Further, reduction in titres over time may be expected to be associated with vaccine failure in those who have received

two doses of vaccine. Trials investigating whether a third dose of vaccine based on the original Wuhan-Hu-1 virus or adapted virus variants will help to prevent symptomatic infection with B.1.617.2 and future virus variants are underway (COV-BOOST https://www.covboost.org.uk/home).

## Supporting information

**S1 Table. Fold reduction in neutralisation by viral variant.** Mean neutralisation of viral variants by DOVE study sera were grouped by vaccine (BNT162b2 or ChAdOx1) and dose (one or two). Fold reduction was calculated by comparing group means. Fold reduction was also calculated from the data after exclusion of the 8/162 samples that possessed N-reactive antibodies by ELISA (Fold (- N +ves)). Significant differences between groups were calculated using One-way ANOVA and Tukey's multiple comparisons test (p values) using GraphPad Prism version 8.
(DOCX)

**S2 Table. Age distribution of DOVE study population.** Median and mean were calculated using Graphpad Prism, descriptive statistics. Age distributions between groups were compared using One-way ANOVA and Tukey's multiple comparisons test, ** p<0.0001.
(DOCX)

**S3 Table. Correlation between neutralising antibody titres against vaccine (Wuhan-hu-1) and VOCs.** Antibody responses measured by pseudotype-based neutralisation assay against Wuhan-hu-1 were compared with those against B.1.617.1, B.1.617.2 and B.1.351. Responses were compared for A) BNT162b2 1 dose, B) BNT162b2 2 doses, C) ChAdOx1 1 dose and D) ChAdOx1 2 doses. Correlations between groups were evaluated and non-parametric Spearman correlation coefficients calculated using GraphPad Prism version 8.
(DOCX)

**S1 Fig. Correlation between antibody titres measured by ELISA and neutralisation assay.** S1 ELISA absorbance 450nm and neutralising antibody titre in the pseudotype-based neutralisation assay against Wuhan-hu-1, B.1.617.1, B.1.617.2 or B.1.351, were compared for each vaccine study group (ChAdOx1 1 dose, ChAdOx1 2 doses, BNT162b2 1 dose and BNT162b2 2 doses). Neutralising antibody titres are expressed as mean (n = 3) +/- SE.
(DOCX)

**S2 Fig. Stratification of vaccine study groups by antibody titre.** Antibody responses measured by pseudotype-based neutralisation assay against Wuhan-hu-1, B.1.617.1, B.1.617.2 and B.1.617.2 were stratified into three groups: titre <50; a titre of 50 to 500; and a titre of >500. The percentage of samples in each group were then plotted for the ChAdOx1 1 dose, ChAdOx1 2 doses, BNT162b2 1 dose and BNT162b2 2 doses.
(DOCX)

**S3 Fig. Correlation between neutralising antibody titre against Wuhan-hu-1 and variants of concern.** Antibody responses measured by pseudotype-based neutralisation assay against Wuhan-hu-1 were compared with those against B.1.617.1, B.1.617.2 and B.1.351. Responses were compared for A) BNT162b2 1 dose, B) BNT162b2 2 doses, C) ChAdOx1 1 dose and D) ChAdOx1 2 doses. Correlations between groups were evaluated and non-parametric Spearman correlation coefficients calculated.
(DOCX)

**S4 Fig. Correlation between neutralising antibody titre and either days post-dose 1 or age at vaccination.** Antibody responses measured by pseudotype-based neutralisation assay against Wuhan-hu-1, B.1.617.1, B.1.617.2 and B.1.617.2 were compared with time post-dose 1 for A) ChAdOx1 and B) BNT612b2, or with age at vaccination for C) ChAdOx1 and D) BNT612b2. E) Age at vaccination for each group (mean +/- SE).
(DOCX)

## Acknowledgments

We are indebted to Therese McSorley for recruiting participants to the DOVE study. We thank all the participants and researchers who have shared genome data openly via the GISAID Initiative.

## Author Contributions

**Conceptualization:** Wendy S. Barclay, Emma C. Thomson, Brian J. Willett.

**Data curation:** William T. Harvey, Emma C. Thomson, Brian J. Willett.

**Formal analysis:** Chris Davis, Nicola Logan, Grace Tyson, Richard Orton, William T. Harvey, David L. Robertson, Emma C. Thomson, Brian J. Willett.

**Funding acquisition:** Wendy S. Barclay, Massimo Palmarini, Emma C. Thomson.

**Investigation:** Nicola Logan, Grace Tyson, Richard Orton, William T. Harvey, Jonathan S. Perkins, Guy Mollett, Rachel M. Blacow, Thomas P. Peacock, Wendy S. Barclay, Peter Cherepanov, Pablo R. Murcia, Arvind H. Patel, David L. Robertson, John Haughney, Emma C. Thomson, Brian J. Willett.

**Methodology:** Richard Orton, William T. Harvey, Jonathan S. Perkins, Guy Mollett, Rachel M. Blacow, Thomas P. Peacock, Wendy S. Barclay, Peter Cherepanov, Massimo Palmarini, Pablo R. Murcia, Arvind H. Patel, David L. Robertson, John Haughney, Emma C. Thomson, Brian J. Willett.

**Project administration:** Chris Davis, John Haughney, Emma C. Thomson.

**Resources:** Wendy S. Barclay, Massimo Palmarini, Emma C. Thomson, Brian J. Willett.

**Supervision:** Jonathan S. Perkins, Guy Mollett, Rachel M. Blacow, Wendy S. Barclay, Massimo Palmarini, David L. Robertson, John Haughney, Emma C. Thomson, Brian J. Willett.

**Validation:** Nicola Logan, Grace Tyson, Brian J. Willett.

**Writing – original draft:** Brian J. Willett.

**Writing – review & editing:** Chris Davis, Richard Orton, William T. Harvey, Jonathan S. Perkins, Guy Mollett, Rachel M. Blacow, Thomas P. Peacock, Wendy S. Barclay, Peter Cherepanov, Massimo Palmarini, Pablo R. Murcia, Arvind H. Patel, David L. Robertson, John Haughney, Emma C. Thomson, Brian J. Willett.

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
