## [Decision Letter · Decision Letter 0]

31 Aug 2021

Dear Professor Thomson,

Thank you very much for submitting your manuscript "Reduced neutralisation of the Delta (B.1.617.2) SARS-CoV-2 variant of concern following vaccination" for consideration at PLOS Pathogens. As with all papers reviewed by the journal, your manuscript was reviewed by members of the editorial board and by several independent reviewers. The reviewers appreciated the attention to an important topic. Based on the reviews, we are likely to accept this manuscript for publication, providing that you modify the manuscript according to the review recommendations.

One of the assigned reviewers ws late and non-responsive and so I read the manuscript myself and agree with the conclusions of the Reviewer #1. I ask the authors address the issue of readability, and also please update the manuscript with references that might have published in the interim. This is also an opportunity to better contextualize what is known about the delta variant and vaccine efficacy, especially with regards to the metric of neutralization  activity that is shown in this study. 

Sincerely,

Benhur Lee

Section Editor

PLOS Pathogens

Benhur Lee

Section Editor

PLOS Pathogens

Kasturi Haldar

Editor-in-Chief

PLOS Pathogens

orcid.org/0000-0001-5065-158X

Michael Malim

Editor-in-Chief

PLOS Pathogens

orcid.org/0000-0002-7699-2064

Reviewer Comments (if any, and for reference):

Reviewer's Responses to Questions

**Part I - Summary**

Reviewer #1: The manuscript “Reduced neutralization of the Delta (B.1.617.2) SARS-CoV-2 variant of concern following vaccination” by C. Davis et al. compares the production of antibodies targeting S1 or RBD of SARS-CoV-2 and the neutralization efficacy of produced antibodies after vaccination with one or two doses of BNT162b2 and ChAdOx1. They show that the BNT162b2 vaccine induces higher antibody production compared to ChAdOx1 after a single dose as well as a double dose. Furthermore, they present reduced neutralizing activity against the mutant viruses, B 1.351, B1.617.1, and B1.617.2 compared to Wuhan-Hu-1 virus after primary and secondary immunization with BNT162b2 or ChAdOx1.

The paper is straight-forward and provides additional information on the effectiveness of vaccination against delta mutations. However, the data is not optimally analyzed, the discussion overlong and repetitive and several references need to be introduced. These issues need to be addressed before it may be accepted.

**Part II – Major Issues: Key Experiments Required for Acceptance**

Reviewer #1: 1) C. Liu also presented the reduced neutralization of the SARS-CoV-2 delta variant after Pfizer and AstraZeneca vaccination. Although the results fit in general to the outcomes in this study, the significantly reduced antibody neutralization ability shown in this study compared to Liu et al (2021), needs to be discussed.

2) A correlation of RBD ELISA data to the in vitro neutralization values with the different pseudotypes would be of value to the readers. It would allow to identify if the loss of neutralization is observed among people with strong antibody responses, or across the board. Likewise, xy scatterplots of Wuhan versus variant(s) neutralization titers would provide more information out of the existing dataset.

4) The discussion is in general overlong and repeats information from the results section. The discussion needs to be shortened, and include references to similar studies (C. Liu et al.), or those that have been published in the meantime since submission. The authors need to discuss the similarities and differences in the phenomena observed in this and other studies. It would be of particular interest to discuss the use of HIV pseudotypes versus VSV ones or of the native SARS CoV-2 in neutralization assays. The existence of other studies should not preclude the publication of this one, but it needs to be contextualized in light of the already available evidence.

**Part III – Minor Issues: Editorial and Data Presentation Modifications**

Reviewer #1: 1) The authors claim that the weaker antibody response of ChAdOx1 compared to BNT162b2 is due to the difference in the age of participants. This claim would be more convincing if the authors would provide the correlation of antibody titers and age. Furthermore, they should cite in this context a paper showing that antibody-producing ability after administration of Pfizer and AstraZeneca vaccines decreases with age (Wall et al. 2021).

2) Line 10 in introduction: mRNA-1273 vaccine (Moderna) is not available from April 2020.

PLOS authors have the option to publish the peer review history of their article (what does this mean?). If published, this will include your full peer review and any attached files.

Reviewer #1: **Yes: **Luka Cicin-Sain

Figure Files:

Data Requirements:

Reproducibility:

References:

---

## [Editor Report · Decision Letter 1]

10 Oct 2021

Dear Professor Thomson,

We are pleased to inform you that your manuscript 'Reduced neutralisation of the Delta (B.1.617.2) SARS-CoV-2 variant of concern following vaccination' has been provisionally accepted for publication in PLOS Pathogens.

Best regards,

Benhur Lee

Section Editor

PLOS Pathogens

Benhur Lee

Section Editor

PLOS Pathogens

Kasturi Haldar

Editor-in-Chief

PLOS Pathogens

orcid.org/0000-0001-5065-158X

Michael Malim

Editor-in-Chief

PLOS Pathogens

orcid.org/0000-0002-7699-2064
---

## [Editor Report · Acceptance letter]

3 Nov 2021

Dear Professor Thomson,

We are delighted to inform you that your manuscript, "Reduced neutralisation of the Delta (B.1.617.2) SARS-CoV-2 variant of concern following vaccination," has been formally accepted for publication in PLOS Pathogens.

Best regards,

Kasturi Haldar

Editor-in-Chief

PLOS Pathogens

orcid.org/0000-0001-5065-158X

Michael Malim

Editor-in-Chief

PLOS Pathogens

orcid.org/0000-0002-7699-2064